# Research in Moral Education: The Contribution of P4C to the Moral Growth of Students

**Félix García-Moriyón [1,*], Jara González-Lamas [2], Juan Botella [3], Javier González Vela [4], Tomás Miranda-Alonso [5], Antonio Palacios [6] and Rafael Robles-Loro [7]**

1   Specific Didactics Department, Faculty of Teacher Training and Education, UAM, 28049 Madrid, Spain
2   School of Education and Health, Universidad Camilo José Cela, 28692 Madrid, Spain; jglamas@ucjc.edu
3   Department of Social Psychology and Methodology, Faculty of Psychology, UAM, 28049 Madrid, Spain; juan.botella@uam.es
4   Physical Sciences, UAM, 28049 Madrid, Spain; javier.gv41@gmail.com
5   Centro de Filosofía para Niños, 28005 Madrid, Spain; tomasmirandaalonso@gmail.com
6   Philosophy Deparment, High School Rayuela, 28935 Mostoles, Spain; anpacib@gmail.com
7   Philosophy Department, High School NSV, 28935 Málaga, Spain; roblesloro@gmail.com
*   Correspondence: felix.garcia@uam.es

**Abstract:** Moral education and moral growth are very important topics, and have been so as much in the fields of moral psychology and moral education as in the policies of governments and international institutions over the past decades. These two topics are also central themes within the educational proposal of Philosophy for Children (P4C), as seen in theoretical reflection and in educational research. It is necessary to start from a more global approach to moral growth, focused on the development of capacities. Such abilities are to be understood as virtues or personality traits that enable us to achieve a full life, that is, to become good people. The transformation of classrooms into communities of philosophical inquiry, following the educational guidelines of P4C, can contribute to the achievement of this objective. Here we present the psychological and methodological underpinnings of an educational research project that we are applying to a small sample that includes two groups—experimental and control—in a typical classroom environment. We are administering tests at the beginning and the end of the application of a moral education model according to the basic principles of Philosophy for Children. The objective is to verify that our research design could be used to evaluate the contribution of this educational model to the students' moral growth, understood as the consolidation of the students' moral habits and competences.

**Keywords:** philosophy for children; moral growth; character; virtues; ethical education; good person; community of philosophical inquiry

## 1. Research Justification

There is currently widespread concern and interest in ethical education, especially within the teaching profession. This concern has been noted by international institutions such as the European Commission [1] and the UNESCO [2]. Normally, this interest has been reflected in the curriculum under various names: Education for Citizenship, Civic and Social Values, Ethical Values, etc. In Spain, for example, the subject that includes ethical education in the curriculum is called "Civic and Social Values" in primary education and "Ethical values" in compulsory secondary education. There are, of course, different approaches and curricula in different countries, but there is general agreement about the need to include moral education as a specific subject in the official curricula.

We understand that compulsory formal education is an institution that transmits moral education to students, mainly through what is often called the hidden curriculum [3–5]. It is not an easy task,

but, in general, students finish their period of schooling with their moral education inculcated either implicitly or explicitly. Obviously, the school is not the only agent for the moral education of the people, but it plays a fundamental role. We also know that there are various proposals for moral education that are supported by different conceptions of ethics, positions that perceive childhood and adolescent moral development differently and which, more importantly, do not share the same ethical objectives. In the contemporary world, pioneering works—by Durkheim [6]), Piaget [7]) or María de Maeztu [8], for example—have made it clear that it is possible and necessary to provide moral education in formal education.

The issue of moral education and the role of the school was taken up in the 1960s. Interest at that time was sparked by a social and cultural crisis, along with a crisis in the educational system itself. Some, but only a few, of these educational systems had already achieved universal schooling by then. Some sectors were especially concerned about the degradation of morals in their societies, and were looking for students to internalize values and behavioral habits. Those positions were part of the character education movement, which had a clearly conservative bias [9]. Other sectors felt the same need, but did not share the same diagnosis. Rather, such groups considered it important to improve moral education in order to move towards democratic societies through critical and progressive pedagogy. Paulo Freire and Matthew Lipman are good representatives of that trend [10] (pp. 53–55), although some scholars emphasize that there are important discrepancies between both [11].

In view of the great existential and global challenges facing humanity [12] in the first decade of the 21st century, along with the demands of consolidating democratic societies—which requires the direct involvement of education [13]—this concern for moral education has increased. There seems to be a consensus—at least in the grand pronouncements—about the need to pay more attention to moral education, as well as to the moral competences (cognitive and affective) and contents (democratic values and human rights) that should be nurtured by teachers and schools.

### 1.1. Research on Moral Development

Achieving an appropriate ethical education involves focusing on the moral growth of students, a broad field of study that has been the topic of numerous theoretical discussions. This can be seen especially in moral and evolutionary psychology, as well as in philosophy and the sociology of education. For decades, the research landscape on moral development has been dominated by Lawrence Kohlberg's approach, which, building on Piaget's theses on moral development, relies heavily on moral reasoning as the basis for ethical behavior. Kohlberg [14] poses six stages of development, each of which is better suited to respond to moral dilemmas. The moral development field has also been heavily influenced by Martin Hoffman, who focused on the affective dimension, with special emphasis on the development of empathy [15,16]. The approach to what we should understand as moral development—which is as much about exploring what we mean by a "morally educated person" or a mature moral agent [17]—has been changing. This can especially be seen in the details of, or in the emphasis placed on, some specific dimension. Yet there are also agreements on some features, which, in addition, have lately expanded with the recent contributions of neurology and more specifically, neuroethics [18,19]. Research in brain development, more specifically at the adolescent stage [20], has provided a particularly interesting contribution. A large amount of information on moral growth has already been attained, although there are some unclear results as well and, above all, a questioning of both the Kohlberg and Hofmann paradigms as frames of reference for research on moral development [21]. We are, therefore, facing a broad field of research, in a stage of change and enrichment, both in approaches and in research methods [22].

The general acceptance and development of character education is of special interest for our proposal. We can leave aside for now the political debate about that approach, a debate that is no longer as intense as it was in the 1980s, although there is still a conservative bias together with the influence of, or confluence with, positive psychology. The latter is a very controversial psychological

trend that proposes a clear set of strengths and virtues that teachers are supposed to foster, or even to instill in students [23].

We focus on defining character as an objective of moral education and on efforts to assess its impact. The most notable aspect of this approach is its emphasis on a set of dispositions and behaviors that define a person, a more global approach than that of Kohlberg and Hoffman [24]. At the present time, with new, more firmly supported contributions and more research on its educational impact, character education shows great theoretical and practical vitality, including the area of evaluation [25]. After an ambitious study in which the impact of character education [26] was not very clear, new approaches are also being developed in this field [27,28] with more promising results in the sense of verifying some improvements in moral development.

In the same way, moral psychology has recently begun to move beyond moral reasoning and to pay attention to the moral self or moral identity. Moral psychologists have embraced the project of investigating the moral self, although researchers differ significantly in their definition of this construct [29]. Examining naturalistic conceptions of moral maturity allow us to go beyond simple themes of principled reasoning, opening up the possibility of analyzing aspects of moral character and virtue that enlarge our understanding of the mature moral agent [17]. Special attention is due here to Darcia Narvaez, an authority in moral development. Her contributions are very valuable, thanks especially to her more comprehensive conception of moral development, one that leads her to propose an integrative model of moral education [30,31]. After moving away from Kohlberg's stage theory to a Neo-Kohlbergian proposal [32,33], she, together with Reilly, highlights the deep relationship between virtues, character and moral education [34], and offers a practical application (along with research) of this proposal [35]. Some recent approaches from the field of psychology support our hypothesis. First, there is a certain consensus around the model of the five major personality factors and even a certain hierarchical organization of those five factors that allow us to talk about a general factor P, also correlated with the Emotional Intelligence (EI) considered as a personality trait [36–38]. This contrasts strongly with another general factor (p) of psychopathy, strongly associated with what is called antisocial personality [39] or also evil, in this case as factor D [40].

On the other hand, psychological research strongly suggests that personality traits predict people's behavior [41,42], leading us to postulate that we can and should take those traits into account when we offer an educational proposal aimed at improving people's moral behavior. This is further reinforced by the malleability of personality traits throughout one's life span, although that malleability decreases with age. It is logical to suppose that said malleability is greater in childhood and in adolescence, but it also occurs in adulthood; at all stages it is possible to intervene in order to cause changes in personality and behavior [43]. It is, however, important to distinguish—whether we speak of cognitive or affective dimensions—between having a high command of rather procedural competences (mastery of argumentation and rhetoric, good manners) and whether or not these dimensions become habits of character or virtues, that is, not only pretending to have them, but actually having them. Some authors even believe that it is the context or the situation that really predicts the behavior, rather than the personality traits, or at least that the situation is complex and we must observe the coherence of the behavior on the one hand and the personality traits and contexts on the other [33,44].

It is not easy, in any case, to establish a close link between personality traits and moral virtues, a task which meets resistance within both the field of moral philosophy and psychology—although the approach is taking root [45–47]. It is also clear that "a number of broad dispositional traits appear to have implications for the moral personality. Certain dispositional profiles—high conscientiousness and agreeableness, and at least moderately high openness to experience—tend to be associated with patterns of behavior and thought indicative of high moral functioning" [48] (p.15). As we explained above, Narvaez and her colleagues' approach is important, precisely because they seek to establish a bridge between an Aristotelian theory of virtues and moral psychology. It is about overcoming a deontological approach to morality, such as Kohlberg's, influenced by Kantian ethics, and a more utilitarian approach, such as that which is found in the work of those who focus moral behavior on

the ability to predict consequences, weighing advantages and disadvantages at the time of making decisions. Aristotelian ethics gives way to an integral understanding of moral growth that takes into account the above aspects, but rather goes towards the achievement of personal fulfillment, or 'floruit' according to classic authors.

*1.2. Philosophy for Children as a Specific Proposal*

The ethical education model of P4C a program that we are applying in the experimental groups of this project, allows us to implement the proposals of character education mentioned above, albeit with a different approach. John Dewey's philosophy had a great influence on the creation of the P4C program and Lipman recognizes Dewey's influence [49]. True education, said Dewey, is that which is given in democracy and for democracy, understanding "democratic society" to mean not only that which has a democratic government, but one in which the relations between people are those of communication. Democracy, he says, is a shared way of life, and human nature can only fully develop in truly democratic groups and societies [50]. Dewey's analysis of human nature and of the relationship between conduct and habits also has a specific influence on the program. Both Matthew Lipman and Ann Sharp [51–54] have a clear approach to ethical education in which the development of the capacity to use the tools of ethical deliberation and decision-making, together with their insistence on the community of inquiry in which behavioral habits are consolidated, tune well with character education [55]. Later authors, such as Daniel, Gregory, Pritchard and Sprod [56–59], provide good evidence of these deep relationships.

Moral growth does not consist of a successive overcoming of stages, but of a process of maturation and progress towards fullness that implies a wider set of moral competences. Therefore, moral education is necessary for children from an early age [60,61]. The P4C educational proposal consists of establishing spaces for reflection about the type of person they want to become and the type of society they want to live in. This communitarian and collaborative reflection emerges from philosophical dialogues in which children and teenagers, together with their teacher, inquire about the moral principles that they consider valuable and that allow them to achieve a good life. To do this, the classroom must become a community of philosophical inquiry in which students and the teacher listen to each other with respect, build their ideas on those of the other participants, question their own beliefs and prejudices in light of the criticism that they mutually contribute, and contrast the arguments with which they support their opinions with the views of others to find good reasons. Engaged in this activity, they develop cognitive and affective skills that are close to what are called cognitive virtues [62].

A community of this type is committed to taking the inquiry beyond the boundaries that limit the field of each discipline, promoting a Socratic dialogue in which students are active subjects of their own learning and are driven to question the problematic aspects of reality and its experience. It is, therefore, an open dialogue on philosophical problems where the classroom is converted into a laboratory of reasonableness, which is necessarily perspectivist but by no means relativistic [63]. One of the characteristics of the deliberation processes in a community of philosophical inquiry is that these deliberations are multifaceted, that is, different interlocutors have different visions, but together they produce a richer, more objective, and less partial vision than that obtained by a single individual from their own point of view.

The behaviors that are generated in a research community have different dimensions: cognitive (development of multidimensional thinking); a dimension related to self-esteem and self-confidence; a socializing dimension, understood as the educational process of self-critical, self-governing and autonomous people; and an ethical dimension, since the dialogue that takes place in the community of inquiry requires—as a necessary precondition—some behaviors based on an attitude of recognition and respect for certain values that are already implicit in every process of sincere dialogue. At the same time, the exercise of dialogue contributes to the development and strengthening of these attitudes and habits, which have a moral character. A general goal of the philosophical research community [64]

can be defined as making the participants become reasonable people rather than rational people, aware that the concept of being reasonable requires a careful and critical analysis to avoid any kind of biases (sexist, cultural, religious . . . ) with specific attention to racist bias, called "white ignorance" or "racialised common sense" [65]. According to Lipman, the term "rationality" is applied to those cognitive processes that have more to do with the method of scientific research and with the disposition of means that allow one to more easily achieve predetermined ends. Reasonability includes a kind of rationality tempered by judgment, which takes into account the problematic aspects of experience, which uses a logic of good reasons, of what is appropriate, of what is correct, while always paying attention to the particular case and the consequences that would derive from a certain action or from maintaining a set of ideas or beliefs. The reasonable person is conscious of the complexities of reality and therefore exercises complex thinking, thinking that is critical, creative and caring [66].

The community of inquiry enables a certain way of acting in the world. It constitutes a means of personal and moral transformation that necessarily leads to a change in the meanings and values that affect the daily actions and judgments of all participants. Of course, even if we think that the Institute for the Advancement of Philosophy with Children (IAPC) curriculum, developed by Sharp and Lipman, is a very sound teaching resource, this is not the only one. More than forty years after the publication of *Harry Stottlemier's Discovery* and the accompanying manual for teachers, *Philosophical Inquiry*, we can find many other teaching resources and different styles of implementation of the community of philosophical inquiry, although the fundamental characteristics of that community, as just mentioned, are shared by most practitioners.

### 1.3. Research in P4C and Moral Development

An important aspect of the P4C educational model is that from the very beginning, it was accompanied by educational research about the impact that the implementation of the program had on student development, although that research focused on the development of cognitive dimensions [67]. The approach taken by Virginia Shipman, who developed a specific test to assess the cognitive skills of the program and collaborated in the evaluation, was an important part of that early research. Following her example, other authors carried out research on the impact of the program, which resulted in a significant amount of positive evidence [68]. Another comprehensive evaluation of the impact of the program can be found in a study done by Trickey and Topping [69] and by a team coordinated by García Moriyón [70]. Both papers provided some clear evidence of the positive impact of the program, especially in the cognitive dimensions, but both indicated that much of the published research did not meet the requirements of educational research. Two more recent meta-analyses reinforce the evidence of the positive impact of the program on the development of cognitive skills [71,72].

A concern for the non-cognitive dimensions of student development has also been evident from the beginning, although with less emphasis. Marie France Daniel's research deals specifically with moral competences. In a quantitative research study, she and her collaborators showed that children in early childhood education significantly improved in four important moral dimensions: autonomy, judgment, empathy, and recognition of emotions [61,73]. Years later, they again verified improvements in these dimensions [74]. Meanwhile, using a qualitative method, Josephine Russell [75] explored the growth of moral conscience with primary students aged 7 to 8. Applying a model of philosophical reflection very close to that of Philosophy for Children, she verified that there was an improvement. Further research has been carried out with older students, and that research has also shown that this improvement is achieved in various social and emotional skills using the model of the Community of Philosophical Inquiry [76,77].

All this indicates that the mastery of competences relevant to moral behavior can be stimulated with the practice of the community of philosophical inquiry. The P4C educational program keeps moral education as a central goal—what Lipman called at the time the development of caring thinking [78]. Ann Sharp [54] maintained that, in addition to developing a set of tools and procedures specific to ethical reflection, another necessity was "the formation of community feelings, which develop the

pro-social virtuous dispositions (such as sincerity, courage, care, honesty, considerateness, compassion, sensitivity, integrity etc.) and character structures of the children in the class". With these words, she clearly linked the difficult task of stimulating mastery of these tools with the community of philosophical inquiry—a space that allows for the ongoing practice of procedures to become a stable mode of behavior. The goal "is to help them (the students) become more thoughtful, more reflective, more considerate, and more reasonable individuals and that, as a result, students will not only have a better sense of when to act but also of when not to act." [79]. Daniel and Auriac [80], Susan Gardner [63] and others, mentioned above, are in the same vein.

Thus, participation in the community of philosophical inquiry helps to forge the character of its participants in such a way that, by increasing self-consciousness, along with self-esteem and empathy, the person builds a moral sensibility, and acts with the conviction that it is possible to build reasonable and just forms of living together. The dialogue also allows agreements to be established concerning values that are universalizables, values that should form the foundation of this type of desirable coexistence. This "moral training" is achieved, in part, thanks to the experience that is produced in the community of inquiry and by establishing in that community relations marked by sincere dialogue, mutual respect, and cooperative research that is free of arbitrariness and manipulation. Through dialogue, two of the fundamental values of a person's moral dimension are also developed: autonomy, linked to the strength of the self; and solidarity, linked to cordiality and pro-sociality. The habits that are generated in the community of philosophical inquiry help the formation and consolidation of moral virtues, which are linked to certain personality traits, in both their cognitive and affective dimensions [55]. They are habits that empower children without limiting their own spontaneity, in a process of constant reconstruction, as proposed by Dewey. [81].

We do, however, lack a structured view of these competences, integrated into a more clearly defined concept of what it might mean to be a good person, understanding moral development in a more comprehensive sense. We set out, therefore, with the presentation of personal competences within a global approach to moral personality [82]. In the process of researching the impact of the program, a detailed analysis of the Institute for the Advancement of Philosophy with Children (IAPC) curriculum made it possible to select a series of personality traits promoted by the program: twenty-three cognitive dimensions and sixteen affective (*ibidem*). The advantage of this approach is that it is framed within a global theory—the theory of multifactorial personality systems elaborated by Royce and Powell [83], used in a heuristic way to organize a certain work project [82] (chap. 5). The following were offered for each dimension: a precise definition, observational variables, evaluation instruments, and intervention strategies in the classroom. In relation to the specific area we are talking about here—moral development—we can pick up some relevant cognitive and affective dimensions. In the cognitive domain: deduction-induction, spontaneous flexibility, sensitivity to problems, fluency of ideas, originality, cognitive complexity, abstract and concrete thinking, etc. The area of affectivity includes, among other aspects, self-feeling, cordiality, realism, cooperation, assertiveness (self-strength), tolerance for the unconventional, reflexivity, achievement motivation and some others.

In the cognitive field, we focus on the ability to perceive the moral dimension of problems and especially on the ability to solve the moral problems we have to cope with. Starting from Kohlberg [84], we believe that solving a moral dilemma or problem is a good test for evaluating a diverse set of cognitive competences—tools that we use to analyze the problem and design appropriate solution strategies while taking into account the values that are at stake [32]. The perception of moral values is important in this type of exercise, along with the possible and necessary hierarchization of them. In the case of conflict, this allows for something intrinsic both to dilemmas and to moral problems—the possibility of deciding to engage in morally good behavior or, sometimes, to opt for the lesser evil or even for the higher good. In part, the resolution of dilemmas is based on the Kantian deontological ethics that served as a starting point for Kohlberg's research. In our model, however, we also consider the assessment of the consequences of our behavior, anticipating the benefits that could be obtained, and also the possible collateral damages. This leads us to utilitarian approaches, which opt at each

specific situation for the lesser evil and put into play what in certain traditions has been called moral discernment—a process that can prove quite complicated [85]. This reworking of the moral dilemma has led us to focus on cognitive dimensions directly related to problem solving.

As we note in the following section, these are fundamentally the dimensions that we are going to evaluate in order to find out whether the transformation of the classroom into a community of philosophical inquiry, according to the model proposed by P4C, would have a significant impact on the moral growth of students.

## 2. A Research Project on Moral Growth of Teenagers

Taking into account the previous literature review, we have proposed a research project, which is underway and should be concluded by the end of 2020. It has two main goals:

1.  To identify and select personality traits that predict virtuous behaviors; that is, peoples' behavior that others would describe as being "good". We have arrived at a provisional concept of what makes a good person based on the current discussion in the fields of moral philosophy, psychology of personality, and moral psychology, as summarized in the previous section [86].
2.  To check whether implementing the P4C moral education model, centered on the transformation of the classroom into a community of philosophical inquiry, improves the moral behavior and/or moral development of teenagers. This requires identifying a set of relevant personality traits that predict good behavior and a specific assessment tool to measure changes in those traits after the educational intervention.

### 2.1. Method

We propose a quasi-experimental design with two groups, experimental and control, carried out in a natural classroom context, with pre-and post-intervention measures. It is a $2 \times 2$ factorial design with one between-subjects factor (two treatment groups) and one within-subjects factor (two measurement times).

All participants are students in Spanish schools. The experimental group involves 85 sixth grade primary education students (11–12 years old), from two private schools, one in Madrid and one in Palencia, and 73 first grade compulsory secondary education students (12–13 years old) from two public secondary schools located in Soto del Real and Móstoles (Community of Madrid). In these groups, teachers qualified for implementing the P4C model are teaching the subjects of Social and Cultural Values in sixth grade and Ethical Values in the first course of compulsory secondary education.

In the control group are 109 first grade compulsory secondary education students (12–13 years old), who are covering the same subjects, but the teaching staff applies different teaching models. The typical model in formal education places more emphasis on knowledge acquisition and significant memorization than on personal critical reflection as in the P4C model. They are in the same high school as the experimental groups, plus a group from two public high schools, one in Madrid and the other in Malaga. The subject in Madrid is taught twice a week in sessions of about 45 min and the other in Málaga once a week.

Before applying the program, we carried out an initial evaluation (pretest) in October–November 2019, and will repeat the assessment at the end of the course, carrying out a post-test evaluation in May 2020.

### 2.2. Dependent Variables

We will evaluate moral development through a set of personality traits or dimensions, both cognitive and affective. The affective dimensions have been selected from those included in the analysis of the application of the P4C program [82].

1.    Cognitive features

    1.1.    S/he argues well and solves problems. S/he uses induction and deduction.
    1.2.    Fluency of written ideas
    1.3.    Originality and moral imagination
    1.4.    Categorical span and conceptual differentiation
    1.5.    Understanding alternative arguments
    1.6.    Conceptual integration

2.    Affective traits

    2.1.    Open mindedness: accepting reasonable criticism and recognizing the value of constructive criticism
    2.2.    Flexibility: welcoming and to another version of a story. S/he is receptive to alternative ideas and evaluates them impartially
    2.3.    Respecting others and their rights. S/he recognizes the right to express one's ideas and does not make negative criticisms directed at people
    2.4.    Self-feeling: S/he is a controlled person with will power, is socially precise, and follows their own image
    2.5.    Agreeableness: showing affection for classmates.
    2.6.    Cooperation: Focusing on others and trusting them
    2.7.    Assertiveness (self-strength): S/he is emotionally stable, mature, grounded in reality, calm, and able to manage emotional difficulties
    2.8.    Tolerance to the unconventional: S/he accepts experiences that do not correspond to one's particular beliefs
    2.9.    Reflexivity versus impulsivity: degree to which the subject considers alternative hypotheses
    2.10.    Achievement motivation: ability to carry out complex tasks in order to achieve certain criteria and maintain them despite adverse conditions that may arise

*2.3. Assessment Tools*

To evaluate cognitive ability, we are using a moral problem [S1-S2] that students had to solve according to the rules previously explained. It is a test developed by one of the authors with a long professional experience in teaching. It is partially based on Kohlberg's original moral dilemma, the standardized and simplified variant by Rest (DIT), and Linz's Konstanz Method of Dilemma Discussion [87,88].

In November 2019 and June 2020, we are using a questionnaire to assess students' behavior, especially that which relates to some of the affective dimensions that appear in the aforementioned research [82], dimensions that we listed in the previous section. This test will be repeated in June 2020. The teacher of each group have to observe the behavior of the students and to fill in the questionnaire [S3]. We are relying also on standard psychometric tests. In this research we are using three.

1.    Personality questionnaire EPQ-J 25′

Developed by H. J. Eysenck and S. B. G. Eysenck, this questionnaire evaluates four basic dimensions of personality: instability or emotionality; extraversion and hardness; and sincerity.

2.    SENA social skills test

This test was created by I. Fernández-Pinto, P. Santamaría, F. Sánchez-Sánchez, M. A. Carrasco and V. Del Barrio, and allows the assessment of the main emotional and behavioral problems of children

and adolescents. It also offers information to evaluate emotional intelligence, adaptation and behavior, and personality.

3.  Self-concept tests AF-5

Developed by García and Musitu, this test evaluates different facets of the self-concept in teenagers: social, academic, emotional, familial, and physical.

*2.4. Data Analysis*

The data recorded with the several tools will be analyzed statistically through linear models (ANOVA), with SPSS. As the participants in the control group could also show a positive or negative change for a variety of reasons, the key to answer our research questions is finding a significant interaction in the expected direction. That is, we expect that as a result of the P4C program the experimental group will change significantly more than the control group, in the expected direction.

The relationships between the moral behaviors and personality factors will be assessed through significance tests on the Pearson correlation coefficients. Finally, we will evaluate the potential moderating role of personality factors in the changes produced in moral behavior as a result of the intervention carried out in the experimental group. We will do this by adding to the ANOVA models a new between-subjects factor when the potential moderator is qualitative (second-order interaction) or through ANCOVA (analysis of covariance) when the potential moderator is a quantitative variable.

## 3. Conclusions

This project is based on an extensive and lengthy research project carried out in the field of psychology of moral development in general and also in the specific field of P4C. Much of the theoretical reflection we have presented in this article, in order to explain and justify our approach, will be used in an analysis of this initial research. Once finished, we will be able to verify whether our approach is adequate and deserves further research.

The underlying problem primarily lies in something that we have already mentioned. The field of moral development research does not have a unified paradigm, rather, it works from different proposals, depending on where the focus is placed. We have opted for a research paradigm that places the development of competences at the center of moral growth. These competences can be identified with virtues in the classical sense of Aristotelian moral philosophy as well as with personality traits used by psychologists. Undoubtedly, these virtues are behavioral habits. There seems to be an agreement that some behavioral habits are closely related to mature moral behavior, that is, with goodness, while others habits (often identified as vices) tend to make mature moral behavior more difficult.

A central problem in this type of research stems from the fact that, when we talk about moral growth and goodness, we are talking about behaviors, not about skills or traits. Beyond the dependable predictive value that certain traits may have, more emphasis needs to be placed on people's behavior, i.e., how they actually tend to behave at different times and in different contexts. This leads us to the awareness of the importance of evaluating behavior not only in the classroom—something that at least initially we have tried with this research—but also in the playground and in family life. This, of course, requires further research. However, what we can discover with our previous work may be very valuable for establishing a good design for future research.

To summarize, we believe that this work offers a suggestive approach based on a comprehensive concept of moral growth, one which links it to the approaches of moral philosophy and psychology that are oriented towards character and virtues, where moral growth is understood as the achievement of a full life, including that of becoming good people [89] (cap.6). The mastery of a set of cognitive and affective competences, in proper balance with each other, where weaknesses in some competences are compensated by strength in others—all of this is what allows us to become good people. This concept of human goodness [86] is well exemplified by the metaphor of the chain and cable, used by Lipman in one of his novels, *Lisa*. In that story, when Millie asks her grandfather if she is inferior to her classmates,

he offers the following metaphor: a chain is as weak as the weakest of its links, while a cable, if it is well braided, remains strong, even if some of its threads are broken. The important thing in becoming a good person is to braid a good cable with the strengths and the weakness that are part of our identity.

**Supplementary Materials:** The following are available online at http://www.mdpi.com/2227-7102/10/4/119/s1, Document [S1]: Dilema-moral-prueba-inicial, [S2] Criterios_para_corregir_problemas_morales, [S3] Observacion destrezas afectivas.

**Author Contributions:** This is a cooperative research project. All the authors have participated in meetings and discussions about the design of the project and all its specific sections. Their contribution was particularly strong in some of the sections, as detailed below: F.G.-M. Coordinator of the project, He has contributed in all the sections of the article, but it was minor in Section 2.4; J.G.-L., Sections 1.1 and 2; J.B., Section 2 with special emphasis in Sections 2.1 and 2.4; J.G.V., Sections 1.2 and 2.2; T.M.-A. Sections 1.2, 1.3, 2.2 and 2.3; A.P. Section 1; R.R.-L., Sections 1 and 2.1. R.R.-L. also coordinates the control groups at the high School in Malaga 7. All authors have read and agreed to the published version of the manuscript.

**Funding:** This research has the institutional and financial support of the IUCE (Insitutto Universitario de Ciencias de la Educacion) UAM.

**Conflicts of Interest:** The authors declare no conflict of interest.

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
