# Peer review of "Research in Moral Education: The Contribution of P4C to the Moral Growth of Students"

_education, doi:10.3390/educsci10040119_

Round 1
Reviewer 1 Report
The author has evidently taken on board several of the comments from my first review, though some appear to have been accepted rather lightly. Overall, the early stages of the article are stronger, though the author continues to make claims that are not evidenced in places. I am confused by the reference to Chetty’s work and racial bias at the beginning of page five as this seems to appear a propos of the need to promote rational and reasonable behaviour. Racism need not feature in this at all. Aside from this element, the section on reasonableness is much improved by the author’s additional section.
As noted previously, the work of Topping and Trickey was not focused on the IAPC curriculum developed by Lipman. This needs to be recorded accurately. In addition, on page five the author notes, ‘Both papers provided some clear evidence of the positive impact of the program, especially in the cognitive dimensions, but both indicated that much of the published research did not meet the requirements of educational research’. This needs to be challenged as Garcia-Moriyon and colleagues determined that research ‘did not meet the requirements of educational research’ because control groups were not always used. It is an extremely narrow view of educational research to suggest that it must involve a control group; indeed, this raises ethical concerns – see my comments from the initial review.
The shift in focus of the article is welcomed. What, instead, has been presented is a more considered piece that sets the scene for a very worthwhile future article. If the author wishes that the study is replicated, thereby lending the original work weight, so additional detail in terms of the methodology will be required here and in the future article. For example, it isn’t quite clear how the features being assessed will be measured. I can, for example, gauge the fluency of writing, though I am less able to measure the respect a participant may have for others or the extent to which arguments are understood. The list of features is very helpful for the reader, but the actual measure is also necessary. The teacher’s observations need to be more clearly articulated; for instance, when and where will these observations take place? How will s/he observe the participants’ behaviour and how will this be recorded? I wonder how the author will ascertain whether any changes might be attributable to the intervention and rule out other factors, for example, that the participants simply mature over the duration of the study. The control group may account for only part of this. In writing about methods it is expected that the author will address and write about ethical consideration in the study.
The conclusion needs a little bit of work because it seems to draw conclusions for which there is no evidence from this particular study. The author needs to be careful not to pre-empt the results and be open to the idea that the intervention may not find what is anticipated or hoped.
The article overall is much improved given its new focus, though the abstract needs to address this. There are some minor errors that will be picked-up in proof-reading. I very much look forward to reading the subsequent article that publishes the results of the study.
Author Response
- I found it interesting in the Research Justification section that values clarification is not noted as part of the background and efforts to address the issue of moral education. While maybe not absolutely essential, in many ways, this popular focus in early classrooms of the 1960s and 1970s in the US at least, added to the confusion in the public schools concerning moral education. VC helped foster the development of home schooling and other alternative models of education to counteract the VC emphasis.
Of course, you're right. Value clarification was very important in the 60s and 70s. However, from our point of view, it is not a current model of moral education. In any case, we don't focus on different proposals of moral education, (so, for example, we don't say anything about Kohlberg's very interesting approach, of transforming schools into just democratic communities). We focus mainly on research on moral growth and on P4C's specific approach to moral education. The mention of value clarification would require to open a new section about models of moral education. And that is impossible at this moment
- Line 117, p.3 the sentence is a bit awkward. I suggest that EI be spelled out in the text and use EI in the parenthetical.
You might be right: “talk about a general factor P, also correlated with Emotional Intelligence (EI) considered as a personality trait”
- It might help to note your proposal more specifically in the beginning since you refer to "our proposal" throughout the document.
Sorry, we don’t know how to tackle with this suggestion I have read again the paper and I think our proposal is very clear in the abstract and in section 2. A research project…, lines 309-320
- Page 5, Line 198: why single out racist bias when it is not specifically discussed earlier? Why not note racist, sexist, cultural and religious biases or all biases? It just seems odd to suddenly in that one sentence to note racism.
We single out “racist bias”, because we are quoting a specific criticism against the implicit racist bias in P4C , However, taking into account your comments, I propose this new writing: a careful and critic analysis to avoid any kind of biases (sexist, cultural, religious…) with specific attention to racist bias, called "white ignorance" or “racialised common sense”” (Chetty, 2018).
- Several sentences are too long and so the meaning is lost.
You might be right, and it may be an effect of the original Spanish style, more familiar with long sentences. The person who reviewed the English of our first text, a professional translator and a person very familiar with philosophy and P4C, paid attention to this problem.
- Line 275, page 6: IAPC is spelled out for the first time but it should be spelled out earlier, the first time you note it. Not later like here.
You are right we move this to line 209: Of course, even if we think that the IAPC (Institute for the Advancement of Philosophy with Children)
- Line 216, page 5: you note the "P4C proposal" but it is actually a model or program. Isn't it better or more correct to say the proposal is based on the P4C model?
You might be right. The most important problem in this case is to be clear that P4C is not a didactic methodology, but a full educational proposal. In any case, we accept: An important aspect of the P4C educational model is that from the very beginning it was accompanied by educational research about the impact
- Verb tense is confusing. At times it is past tense but others present or future tense. The proposal work began in 2019 so that is past tense. But it extends into 2020 and that is present tense.
I'll check the text. In any case, you acknowledge that this article is about a past project and an ongoing research.
- Overall, I would like to see more discussion or explanation of the project/proposal. Otherwise, it seems more like a review of moral education with a brief look at P4C and I feel that more discussion of the project/proposal you are working on needs to be brought out more to connect with the discussion of moral education that you offer. We know a bit more this time about the groups formed but it needs more discussion.
You are right that this article should be followed by a second article in which we will discuss the results of the ongoing research. That is, it requires a second article not to extend this.
- Conclusion appears to be done hastily. Not only many typos but it is not clear and cohesive and does not do justice to the article. Also, you note Lipman's story but it is actually a novel. Big difference.
Yes, it is a novel written by Lipman, not a story. I do not know whether “hastily” is the right word. Maybe is a short conclusion and we can extend it a little more.
Reviewer 2 Report
- I found it interesting in the Research Justification section that values clarification is not noted as part of the background and efforts to address the issue of moral education. While maybe not absolutely essential, in many ways, this popular focus in early classrooms of the 1960s and 1970s in the US at least, added to the confusion in the public schools concerning moral education. VC helped foster the development of home schooling and other alternative models of education to counteract the VC emphasis.
- Line 117, p.3 the sentence is a bit awkward. I suggest that EI be spelled out in the text and use EI in the parenthetical.
- It might help to note your proposal more specifically in the beginning since you refer to "our proposal" throughout the document.
- Page 5, Line 198: why single out racist bias when it is not specifically discussed earlier? Why not note racist, sexist, cultural and religious biases or all biases? It just seems odd to suddenly in that one sentence to note racism.
- Several sentences are too long and so the meaning is lost.
- Line 275, page 6: IAPC is spelled out for the first time but it should be spelled out earlier, the first time you note it. Not later like here.
- Line 216, page 5: you note the "P4C proposal" but it is actually a model or program. Isn't it better or more correct to say the proposal is based on the P4C model?
- Verb tense is confusing. At times it is past tense but others present or future tense. The proposal work began in 2019 so that is past tense. But it extends into 2020 and that is present tense.
- Overall, I would like to see more discussion or explanation of the project/proposal. Otherwise, it seems more like a review of moral education with a brief look at P4C and I feel that more discussion of the project/proposal you are working on needs to be brought out more to connect with the discussion of moral education that you offer.We know a bit more this time about the groups formed but it needs more discussion.
- Conclusion appears to be done hastily. Not only many typos but it is not clear and cohesive and does not do justice to the article. Also, you note Lipman's story but it is actually a novel. Big difference.
Author Response
This second draft is greatly improved.
I offer minor editorial suggestions only.
I suggest deleting the sentence (lines 7 to 9) that reads: The moral growth of students...over that time." The sentence disrupts the flow of thought that begins in the sentence before it and that then continues in the sentence after it. The sentence is also not needed. The sentence that comes before this states clearly that moral growth has been studied by moral psychologists.
We agree. New sentence: Moral education and moral growth are very important topics, as much in the fields of moral psychology and moral education as in the policies of governments and international institutions over the past decades.
As suggest line 9 be edited to read: "These two topics are also central themes within." The suggested edit reverses the order of the words "topics" and "themes." Because the word "topics" is used in in the passage that comes before this sentence, the reader can follow more easily if the word "topics" is used in this sentence to refer to the same thing as in the preceding passage before a new terms for the same thing is introduced.
We agree: These two topics are also central themes within the educational proposal of Philosophy for Children, as seen in theoretical reflection and in educational
In line 44 the phrase "in Spain" could be misleading. The phrase refers to Maria de Maeztu whose research was done in Spain. However, a reader who is not familiar with the sources cited could think that the phrase "in Spain" refers not only to de Maeztu but also to Durkheim and Piaget. To avoid possible misunderstanding the authors might consider deleting the phrase "in Spain." If they think it is necessary to specify de Maeztu's national background, they could write "or the Spanish researcher Maria de Maestu."
We agree: In the contemporary world, pioneering works - by Durkheim (2002, [1922]), Piaget (1984, [1934]) or María de Maeztu (1938) for example -
In line 56 when the authors write "discrepancies between both" do they mean "differences between the two"?
Sorry, we do not see any difference between “both” and “the two”. However, “discrepancies” is more appropriate than “differences” in this context.
The statement about Kohlberg in line 144 is confusing. I think this confusion could be cleared up if the word "offered" is substituted for the word "did." The author is contrasting the approach developed with that of Narvaez et al., who sought to overcome a purely deontoloigical approach by developing a theory that incorporates insights from neuroscience and evolutionary theory. It seems to me that what the authors are trying to say in the line in question is that "It (the approach of Narvaez et al.) is about overcoming a deontological approach, such as Kohlberg...offered."
We agree: It is about overcoming a deontological approach to morality, such as Kohlberg, influenced by Kantian ethics, offered.
In line 407 I suggest "extensive and lengthy research" in place of "extensive long research."
We agree; you might be right: This project is based on an extensive and lengthy research project
In line 410, "will e will be" should be just "will be."
Agree: will be used in an analysis of this initial research
In line 436 and following I suggest that the sentence that begins "Millie's grandfather..." be replaced with "In that story, when Millie asks her grandfather if she is inferior to her classmates, he offers the following metaphor: a chain is as weak as the weakest of its links, while a cable, if it is well braided, remains strong." -- The suggested rewrite clears up a few editorial problems with the passage as written. It also presents the content of the sentence in a way that is easier for the reader to follow. By replacing the word "novel" with "story" there is a clearer connection with the sentence that comes before it. By moving the reference to "story" to the beginning of the sentence, there is a smoother transition of thought from the previous sentence to this sentence. By moving the word "metaphor" to just before the metaphor is offered the sentence is easier to follow, and the most important elements of the sentence, namely the word "metaphor" and the metaphor itself are both at the end of the sentence. If the two most important elements are together at the end of the sentence, they are more likely to be remembered.
We agree, your sentence is easier for the reader to follow; however, in the website of the IPAC call “novel” to the stories of the curriculum: used by Lipman in one of his novels, Lisa. In that story, when Millie asks her grandfather if she is inferior to her classmates, he offers the following metaphor: a chain is as weak as the weakest of its links, while a cable, if it is well braided, remains strong, even if some of its threads are broken."
Reviewer 3 Report
This second draft is greatly improved.
I offer minor editorial suggestions only.
I suggest deleting the sentence (lines 7 to 9) that reads: The moral growth of students...over that time." The sentence disrupts the flow of thought that begins in the sentence before it and that then continues in the sentence after it. The sentence is also not needed. The sentence that comes before this states clearly that moral growth has been studied by moral psychologists.
As suggest line 9 be edited to read: "These two topics are also central themes within." The suggested edit reverses the order of the words "topics" and "themes." Because the word "topics" is used in in the passage that comes before this sentence, the reader can follow more easily if the word "topics" is used in this sentence to refer to the same thing as in the preceding passage before a new terms for the same thing is introduced.
In line 44 the phrase "in Spain" could be misleading. The phrase refers to Maria de Maeztu whose research was done in Spain. However, a reader who is not familiar with the sources cited could think that the phrase "in Spain" refers not only to de Maeztu but also to Durkheim and Piaget. To avoid possible misunderstanding the authors might consider deleting the phrase "in Spain." If they think it is necessary to specify de Maeztu's national background, they could write "or the Spanish researcher Maria de Maestu."
In line 56 when the authors write "discrepancies between both" do they mean "differences between the two"?
The statement about Kohlberg in line 144 is confusing. I think this confusion could be cleared up if the word "offered" is substituted for the word "did." The author are contrasting the approach developed with that of Narvaez et al., who sought to overcome a purely deontoloigical approach by developing a theory that incorporates insights from neuroscience and evolutionary theory. It seems to me that what the authors are trying to say in the line in question is that "It (the approach of Narvaez et al.) is about overcoming a deontological approach, such as Kohlberg...offered."
In line 407 I suggest "extensive and lengthy research" in place of "extensive long research."
In line 410, "will be will be" should be just "will be."
In line 436 and following I suggest that the sentence that begins "Millie's grandfather..." be replaced with "In that story, when Millie asks her grandfather if she is inferior to her classmates, he offers the following metaphor: a chain is as weak as the weakest of its links, while a cable, if it is well braided, remains strong." -- The suggested rewrite clears up a few editorial problems with the passage as written. It also presents the content of the sentence in a way that is easier for the reader to follow. By replacing the word "novel" with "story" there is a clearer connection with the sentence that comes before it. By moving the reference to "story" to the beginning of the sentence, there is a smoother transition of thought from the previous sentence to this sentence. By moving the word "metaphor" to just before the metaphor is offered the sentence is easier to follow, and the most important elements of the sentence, namely the word "metaphor" and the metaphor itself are both at the end of the sentence. If the two most important elements are together at the end of the sentence, they are more likely to be remembered.
Author Response
The author has evidently taken on board several of the comments from my first review, though some appear to have been accepted rather lightly. Overall, the early stages of the article are stronger, though the author continues to make claims that are not evidenced in places. I am confused by the reference to Chetty’s work and racial bias at the beginning of page five as this seems to appear a propos of the need to promote rational and reasonable behaviour. Racism need not feature in this at all. Aside from this element, the section on reasonableness is much improved by the author’s additional section.
We mention “racist bias”, because we are quoting a specific criticism against the implicit racist bias in P4C , However, taking into account your comments, I propose this new writing: a careful and critic analysis to avoid any kind of biases (sexist, cultural, religious…) with specific attention to racist bias, called "white ignorance" or “racialised common sense”” (Chetty, 2018).
As noted previously, the work of Topping and Trickey was not focused on the IAPC curriculum developed by Lipman. This needs to be recorded accurately. In addition, on page five the author notes, ‘Both papers provided some clear evidence of the positive impact of the program, especially in the cognitive dimensions, but both indicated that much of the published research did not meet the requirements of educational research’. This needs to be challenged as Garcia-Moriyon and colleagues determined that research ‘did not meet the requirements of educational research’ because control groups were not always used. It is an extremely narrow view of educational research to suggest that it must involve a control group; indeed, this raises ethical concerns – see my comments from the initial review.
About Topping and Trickey; of course they do not implement IAPC curriculum. So, we say in lines 237-241 that they, and also Claire Cassidy: "Applying a model of philosophical reflection very close to that of Philosophy for Children, she verified that there was an improvement. Further research has been carried out with older students, and that research has also shown that this improvement is achieved in various social and emotional skills using the model of the Community of Philosophical Inquiry»
On the statement "did not meet the requirements of educational research", of course anyone could question this statement, but we quote two articles and both justify this criticism; therefore, these are the articles that should be consulted. Their criticism is based on the fact that many of the articles analyzed for the meta-analysis do not "provide the appropriate data: means, standard deviations, and number of participants, for both pre-test and post-test measurements. These data are necessary to facilitate possible reanalyses such as the one performed in this article"..
The shift in focus of the article is welcomed. What, instead, has been presented is a more considered piece that sets the scene for a very worthwhile future article. If the author wishes that the study is replicated, thereby lending the original work weight, so additional detail in terms of the methodology will be required here and in the future article. For example, it isn’t quite clear how the features being assessed will be measured. I can, for example, gauge the fluency of writing, though I am less able to measure the respect a participant may have for others or the extent to which arguments are understood. The list of features is very helpful for the reader, but the actual measure is also necessary. The teacher’s observations need to be more clearly articulated; for instance, when and where will these observations take place? How will s/he observe the participants’ behaviour and how will this be recorded? I wonder how the author will ascertain whether any changes might be attributable to the intervention and rule out other factors, for example, that the participants simply mature over the duration of the study. The control group may account for only part of this. In writing about methods it is expected that the author will address and write about ethical consideration in the study.
We very much appreciate these comments, because they point to the basic requirements of our research. However, we think this is not the time to do so. We are working on the research, and all of this information will be shared in a specific document once the research is finished.
I think we are providing the basic information in this paper (2.3. and 2.4) . We night share, as appendix to this rticle) the two documents: the moral problem and the questionnaire to assess students’ behavior
The conclusion needs a little bit of work because it seems to draw conclusions for which there is no evidence from this particular study. The author needs to be careful not to pre-empt the results and be open to the idea that the intervention may not find what is anticipated or hoped.
You might be right, but we think that the conclusion is very prudent: “we believe that this work offers a suggestive approach…”. We are more assertive with the theoretical hypothesis: “The mastery of a set of cognitive and affective competences, in proper balance with each other, where weaknesses in some competences are compensated by strength in others - all of this is, is what allows us to become good people”, but most of the paper explores and tries to justify this hypothesis. The next research might help us to move forward in our most important goal: to foster moral growth of students in such a way they can become good people.
The article overall is much improved given its new focus, though the abstract needs to address this. There are some minor errors that will be picked-up in proof-reading. I very much look forward to reading the subsequent article that publishes the results of the study.
Thanks for your positive evaluation. We have read again the initial abstract and we can’t find out how to improve our presentation of the paper.